# Development and validation of the DHIS2 platform for integrating sociomedical data to study wound care outcomes

**Atika Rahman Paddo**[1]*, **Snigdha Kodela**[1], **Lava Timsina**[2], **Shomita S. Mathew-Steiner**[3], **Saptarshi Purkayastha**[1]*, **Chandan K. Sen**[3]

**1** Department of BioHealth Informatics, Indiana University - Purdue University Indianapolis, Indianapolis, Indiana, United States of America, **2** Center for Outcomes Research, Department of Surgery, Indiana University School of Medicine, Indianapolis, Indiana, United States of America, **3** Indiana Center for Regenerative Medicine and Engineering, Department of Surgery, Indiana University School of Medicine, Indianapolis, Indiana, United States of America

* apaddo@iu.edu (ARP); saptpurk@iu.edu (SP)

## Abstract

Wound trajectory and outcomes research has applications in different aspects of wound healing: forecasting wound healing time, access and utilization of wound care services, factors associated with disparities in wound care services, and its quality and outcomes. Wound care research benefits from a well-maintained record management system. In this article, we demonstrate the customization of the District Health Information Software (DHIS2) platform to integrate wound care clinical data with social determinants of health from several Comprehensive Wound Centers (CWC) in Indiana. We describe the modules and features of our platform, such as tracker capture, visualization, and maps. DHIS2 is used in more than 60 countries to monitor and evaluate health programs. However, to the best of our knowledge, this is the first attempt to use DHIS2 as a wound care data warehouse, a platform to perform wound care research for academic researchers and clinical practitioners. Clinicians can use the platform as one of the key tools to make an informed decision in determining the treatment for favorable healing trajectory and wound outcomes. We conducted a usability and acceptance survey among researchers at the Indiana Center for Regenerative Medicine and Engineering and found that DHIS2 can be a suitable infrastructure to manage metadata to import and analyze combined data from disparate sources, including Electronic Medical Records, WoundExpert, and clinical trials management software like REDCap.

## 1 Introduction

The development of systematic and continuous maintenance of electronic health records (EHR) for wound care services and outcomes helps the clinician, medical staff, or caregiver to plan and provide appropriate wound treatments and services to patients with chronic wounds requiring long-term care and follow-up [1]. A wound care database that tracks wound

**Data Availability Statement:** The survey data from the participants for the DHIS2 WoundInfo platform evaluation is shared after deidentification (IRB #16347) as supplementary material with the paper.

However, due to ethical-legal considerations, clinical data cannot be shared publicly as it contains protected health information (PHI) and consent was not sought from study participants for public data sharing. Data is available from the Indiana University Research Electronic Data Capture (REDCap) data governance committee for researchers who meet the criteria for access to confidential data. Data governance committee REDCap, Indiana University, email: redcap@iu.edu and Human Subjects and Institutional Review Board of Indiana University, email: irb@iu.edu.

**Funding:** The author(s) received no specific funding for this work.

**Competing interests:** The authors have declared that no competing interests exist.

trajectory in relation to treatment enables informed decisions, increases service efficiency and effectiveness, and helps monitor difficult-to-heal chronic wounds to prevent long-term complications [2, 3]. Patient characteristics, including clinical measures, socio-economic conditions, and demographics, referred to as sociomedical data [4], can help predict wound trajectory. Wounds not healing for 2 weeks or completely within 6 weeks are considered non-healing and require specialized treatment [5, 6]. Typical chronic wounds such as diabetic foot ulcers and venous leg ulcers have brought substantial difficulties to millions of patients worldwide [7]. Indiana's poor wound care outcomes, ranked among the bottom ten states in the US, are largely due to complications in managing difficult-to-heal chronic diabetic or traumatic wounds, including challenges with wound size measurement, assessment, healing monitoring, and case management [8]. Western Indiana has higher amputation rates, indicating significant geographic disparities [9]. A health management information system (HMIS) maybe used as a chronic wound database to identify clusters of chronic wounds and complications based on patient address geolocation. This database can also monitor and evaluate wound care services, improving efficiency and patient outcomes. The IU Health Comprehensive Wound Center (IUH-CWC) uses HMIS to integrate geographical, social, economic, and medical data from several clinical and observational studies to analyze chronic wounds' distribution and determinants and provide evidence-based data for proper wound management in Indiana. IUH-CWC is also one of the four national clinical research units of the NIH-funded Diabetic Foot Consortium (DFC), a national network to study diabetic foot ulcers/wounds and their complications [10].

District Health Information Software Version 2 (DHIS2) is an open-source, web-based HMIS platform for data collection, management, and analysis [11]. It is the de-facto standard for managing health resources for monitoring and evaluating health programs in over 65 countries, from the clinic to the national level. Clinicians can use DHIS2 to monitor patient health, improve disease surveillance, map disease outbreaks, and speed up health data access for health facilities and government organizations. We used the DHIS2 platform to store and analyze the data we received from CWC, focusing on precisely wound care and services data from several wound-related clinical studies in Indiana.

The study objective was to customize DHIS2 to create a wound data analysis platform— WoundInfo, and present its development and validation process so that other wound care programs can learn from our experience. Although no previous projects have used DHIS2 as a wound care data warehouse, we explored prior published research in data integration to develop metadata and data modeling in DHIS2.

## 1.1 DHIS2 as a health data warehouse

Previous research on resource allocation (human, natural, and capital) and data management through DHIS2 has identified its efficacy as a health program integration platform. However, there is limited or no previous published work on using DHIS2 for chronic wounds. Work by Jayatilleke et al. [12] discussed injury surveillance through DHIS2 for several patients in Sri Lanka, and further, customization and piloting DHIS2 in a resource-constrained setting to perform injury surveillance and identify appropriate interventions by Jayatilleke et al. [13]. These customizations have been broadly categorized by Saunders et al. [14] as low-resource settings research that expands routine health services monitoring and evaluation into a new clinical domain. Researchers in Guinea also utilized DHIS2 for real-time disease surveillance in West Africa [15]. This research builds upon previous work by integrating data from multiple clinical studies, electronic health records (EHRs), and socio-economic data sourced from the United States Census and other relevant databases. The incorporated clinical studies are

registered on ClinicalTrials.gov under the following identifiers: NCT02577120, NCT02581098, NCT03793062, and NCT01101854.

To learn about other methods of integrating disparate data systems in health care, we examined work by Marongwe et al. [16], who attempted to verify the role of active surveillance in monitoring adverse events in Zimbabwe. Faujdar et al. [17] evaluated ICT-based health information systems in India, even though they highlighted that the long-term sustainability and integration of the systems remain a challenge. Kariuki et al. [18] also explain the challenges of implementing interoperability across health information systems and discuss automating indicator data reporting from EMR to aggregate data using Open Medical Record System (OpenMRS) and DHIS2. Nawaz et al. [19] performed an interventional study with an assessment of DHIS and aggregating multiple programs in it, such as the Expanded Program on Immunization (EPI), Lady Health Worker (LHW) program, and assessed the improvement in HIS in the districts of Pakistan. Purkayastha et al. [20] analyzed how DHIS2 and its related model could perform analytics on big data integrated from various data sources, which shows the potential of expanding our current work to wound centers all over the nation.

Other useful software and applications can capture structured wound data from DHIS2 or other electronic methods in a helpful fashion, such as Intellicure [21]. Intellicure has partnered with US Wound Registry [22] for clinical research and data analytics that lead to improved outcomes for patients with chronic wounds.

## 2 Materials and methods

Our project's initial plan was to identify high-risk patients (Diabetic patients, patients with poor circulation, or patients having peripheral vascular diseases, venous insufficiency, Rheumatoid arthritis, and so forth), track the readmissions, and forecast the patient load by geographies, seasonality, and comorbidities. During the process, the project focused more on data analysis, visualization, and forecasting based on demographic parameters like gender, race, ethnicity, wound healing duration, or wound area reduction across geographies. Several steps were taken during the process:

1. Deploy a DHIS2 development instance,

2. Initialize the metadata of the DHIS2 instance,

3. Data import,

4. Data synchronization

### 2.1 Deployment of DHIS2 instance

At the beginning of the project, we created a GitHub repository to monitor the progress and document all our work. First, we started with the deployment process. We deployed DHIS2 to a development server. For this, we used the domain https://woundinfo.us, and used virtual machines hosted on the XSEDE High-performance Computing Infrastructure. We deployed DHIS2 on a PostgreSQL database with Tomcat as the Java Application Server in Ubuntu. We used a reverse proxy to hide the ports, install an SSL certificate, and perform caching to support 1000s of simultaneous users. The official DHIS2 documentation were followed during installation and customization [23–27].

Simultaneously, while deploying the instance, we created a combined data dictionary for the data from the six individual wound-related clinical studies provided by CWC. CWC uses

REDCap, a secure web-based electronic data capture platform for collecting and managing all clinical trials and surveys [28, 29].

## 2.2 Initialization of the DHIS2 instance

Once we completed the deployment process, we worked on customizing DHIS2 for Indiana and the wound care clinical domain. We learned from existing work going on to create a state-wide wound care registry [30] when designing WoundInfo based on DHIS2. We added Organization units in the DHIS2 platform as State, Regions, and Counties and their respective zip codes. After this, we implemented Geographic Information System (GIS) configurations with shapefiles of Indiana maps, including the zip codes. A Common Data Dictionary between WoundExpert and REDCap was created simultaneously. We created Data elements, Option sets & Categories for Demographics form along with Wound Care Program and Registration Program stage in DHIS2. We posted tracked entity instances, enrollment, and event items into the instance. We posted the data values through Python, and the process was automated to create/update data values for the Demographic form through Python code.

## 2.3 Data import

In the next step, we imported metadata and data values from multiple wound-related clinical studies that capture wound history, study visits information, wound healing status, patient demographics, and the zip codes for the patient's residence. In DHIS2, final dashboards and visualizations were developed, including charts, graphs, and maps. We followed statistical modeling (SES) for the essential attributes and later created a platform design report. We performed stratified analysis on the data to see the significant differences between the independent groups using a one-way ANOVA test on healing and non-healing patients of different races and genders, followed by posthoc Mann-Whitney U test for the differences of the visit data among the patients as well. Finally, we designed instruments for user feedback and created a platform design report with each iteration so that formal user feedback could be captured. Along with this, we prepared a questionnaire for data visualization and modeling.

We created a strategy for the next phase while implementing the feedback instruments and analyzing user feedback. We obtained data on socio-economic status at the zip-code level from Indiana University's Polis Center, a respected national center for social and behavioral data. The Polis Center is a unit in the IU School of Informatics and Computing that works with community partners in Indiana and experts in the Polis Center use geospatial technologies to integrate, manage, and visualize the rapidly growing information on several places and analyze local, state, and national data sources to understand social issues and how they impact communities [31]. Visuals for all wound-related projects were imported for the inputs simultaneously. IU CWC also uses two types of EHR for clinical documentation of wound patients—WoundExpert and Cerner. We integrated metadata for wound assessments from WoundExpert data format for each tracked entity instance (i.e., patient wound). Based on the zip codes, we also imported geographical census-level data for the years 2019–2021 from the Polis Center into DHIS 2.

Visualizations were created from those data in the DHIS2. For all the dictionaries, we updated data visualizations based on the feedback from time to time. Instruments for user input were created while we gathered and made modifications. Simultaneously, user manuals for our WoundInfo platform were created based on the DHIS2 user documentation.

## 2.4 Maps configuration

For the GIS configuration of the instance, we obtained full dataset shapefile for Zipcode number and Objectid, downloaded from https://www.IN.gov/. These obtained shapefiles were converted to GML (Geography Markup Language) format [32]. The GML files were prepared after transforming from EPSG 26916 to EPSG 4326 format. The GML files were updated for 4 levels of organization units in DHIS2 including both county and zip code levels. The ZCTA level maps were obtained from official https://www.census.gov/ site. Then we had to match the coordinates of the zip codes of the organization units in the Maps module with the zip code coordinates obtained from the imported research study data. For that, we mapped the UIDs of the zip codes and updated the data values again to match them with the Maps' coordinates.

## 2.5 Data synchronization

To make the platform more accessible for forecasting purposes, we created a "Parent Program" where all wound care data from all the wound-related clinical studies were included. This Parent Program has access to every patient's wound characteristics and socio-demographic data, including every visit and sample collection information for the patients. Every patient was brought into the same type of identification numbers which eased the scaling of the imported data. Finally, we scaled the system by importing from all data systems. The tracker capture report for individual patient can be viewed in Fig 1.

## 2.6 Feedback on the usability of DHIS2 platform

In collaboration with the Indiana Center for Regenerative Engineering & Medicine (ICRME), we performed a study (IU IRB #16347) to get usability feedback from researchers who use our platform. The recruitment period of this study was from September 26, 2022 to November 15, 2022. The informed consent from the participants was obtained through email. This study provided them with a user-friendly data analysis and visualization platform. This platform provided them access to different modules designed for—generating pivot tables, visualizations,

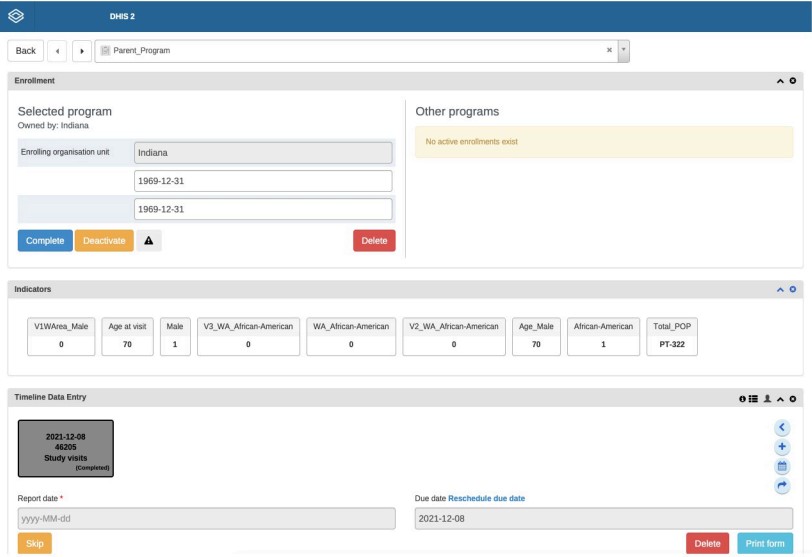

**Fig 1. Tracker capture report of parent program.** The interface of the Tracker Capture Report for the program named **Parent program** from the DHIS2 instance.

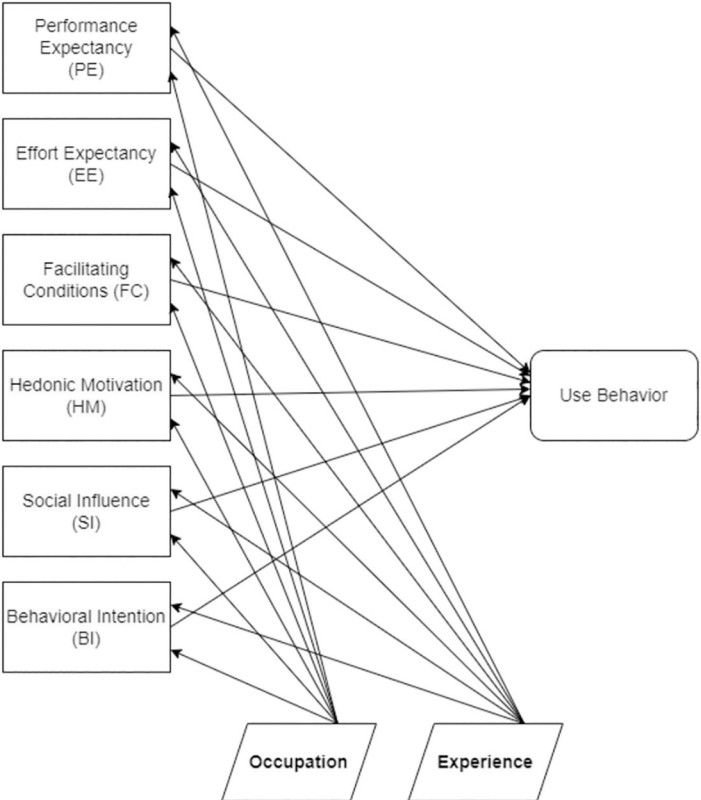

**Fig 2. UTAUT2 model constructs to determine use behavior.** The model diagram of UTAUT2 constructs to determine the use behavior of survey participants.

maps, and standard reports from their analysis. Users also created their own dashboards to get an updated, single-shot view of their data and analysis. Their participation in the study and feedback helped us plan this platform's improvements. 12 researchers aged 18 or older years and either working in or collaborating with ICRME participated in the study.

We used 6 constructs from the second generation of the Unified Theory of Acceptance and Use of Technology (UTAUT2) model as a basis for the survey questionnaire. Generally, this model can be used to investigate the effects of different constructs on the usability of products and platforms. Tamilmani et al. [33] studied and explained technology adoption through the systematic review of UTAUT2 literature. Schomakers et al. [34] used a UTAUT2 model to explain user acceptance of a lifestyle and therapy mobile health app, while Palas et al. [35] used UTAUT2 constructs to find the acceptance of mobile health services by elderly people. We mapped our survey questions to the UTAUT2 constructs: Performance Expectancy (PE), Effort Expectancy (EE), Facilitating Conditions (FC), Hedonic Motivation (HM), Social Influence (SI), and Behavioural Intention (BI); showing the UTAUT2 model in Fig 2.

## 3 Results

The visualizations and analysis using the built-in DHIS2 apps allowed us to analyze the patient distributions across different clinical trials and draw out the relation and pattern of patient distribution in several categories. We found out that out of 277 patients registered in six trials, 44.2% are females, and 55.8% are male patients. The racial distribution is Caucasian 74.3%,

African American 23.9%, Asian 1.1%, and Other races 0.72%. The wound healing status of the population by the end of the first (n = 187), second (n = 168), and third (n = 68) visits are 'healing' in 44.9%, 64%, and 50%, respectively. The final wound healing status (n = 121) is healed in 72.7% of the patients and not healed in 27.3% of the patients. The average time of follow-up (indicator showing the difference in first and final visit dates) for the completely healed wounds over the years 2019–21 are 23.2 weeks, 13.3 weeks, and 3.1 weeks while, for the non-healed wounds, it is 15 weeks, 14.8 weeks and 8.5 weeks, respectively. We conducted a stratified analysis and found that these differences in first and final visit dates for healed and non-healed wounds are statistically insignificant, with a p-value of 0.3386.

## 3.1 Race and gender distribution

The race distribution of patients with healing wounds (n = 84) at the first visit is Caucasians 81.5%, African-Americans 18.5%. Non-healing wounds (n = 102) are comparable in Caucasians (74.7%), and African-Americans (25.3%). The gender distribution of patients with healing wounds (n = 84) is females 51%, males 49%. Non-healing wounds (n = 103) are comparable in females (43.7%) and males (56.3%). Over the 3 years (2019, 2020, 2021), the number of patients with healing wounds at the first visit is 9, 45, and 12, respectively, for Caucasian patients, while 4, 9, and 2 for African-American patients. We can see the patient distribution by year from the following cross-tabulation in Table 1.

Terlizzi et al. [36] in National Health Statistics Reports 2019 states that estimates by race and Hispanic ethnicity, sex, age among US were compared for statistically significant differences using two-tailed tests having critical value for significance as 1.96. Here we get a p-value of 0.201 in statistical analysis (One-way ANOVA test) for the above race and gender distribution, which implies that the differences in different density groups for healing and non-healing patients are not statistically significant.

## 3.2 Wound area distribution

Time-dependent change in wound area is one of the critical measurements to determine the chronicity of the wounds. The average wound area (sq. cm) for 2019 in males is 43.4; in females, it is 16.1, while the average (n = 322 total wounds in all three years- 2019, 2020 & 2021) is 32.5 sq. cm. In 2020, the values were 47.6 in males and 21.5 sq. cm in females, while the average was 32.8 sq. cm. In 2021, the values were 47.8 in males and 33.8 sq. cm in females, while the average was 44.3 sq. cm. For all three years combined, the average wound area values

**Table 1. Race and gender distribution over the years.**

| Race | Caucasian Heal. | | Caucasian Not heal. | | African-American Heal. | | African-American Not heal. | |
|------|---------|---------|---------|---------|---------|---------|---------|---------|
| | Patient | Percent | Patient | Percent | Patient | Percent | Patient | Percent |
| 2019 | 9 | 18.75% | 27 | 56.25% | 4 | 8.33% | 8 | 16.67% |
| 2020 | 45 | 46.88% | 32 | 33.33% | 9 | 9.38% | 10 | 10.42% |
| 2021 | 12 | 40% | 11 | 36.67% | 2 | 6.67% | 5 | 16.67% |
| Gender | Female Heal. | | Female Not heal. | | Male Heal. | | Male Not heal. | |
| | Patient | Percent | Patient | Percent | Patient | Percent | Patient | Percent |
| 2019 | 6 | 12.77% | 19 | 40.43% | 7 | 14.89% | 15 | 31.91% |
| 2020 | 14 | 24.14% | 14 | 24.14% | 9 | 15.52% | 21 | 36.21% |
| 2021 | 5 | 23.81% | 0 | 0% | 8 | 38.10% | 8 | 38.10% |

**Table 2. Average wound area distribution over the visits.**

| Attributes | Visit1 Wound Area (sq.cm) | Visit2 Wound Area (sq.cm) | Visit3 Wound Area (sq.cm) |
|---|---|---|---|
| *Males* | 46.3 | 31.1 | 13.3 |
| *Females* | 22.1 | 13.1 | 5.2 |
| *Caucasians* | 32.7 | 21.6 | 15.6 |
| *African – Americans* | 42.7 | 32.1 | 12.2 |
| *Diabetics* | 25.9 | 18.4 | 14.4 |

for male and female patients were 46.3 sq. cm and 22.1 sq. cm respectively. We compared the values in Caucasian (2019–22.6, 2020–28.8, 2021–49.9) and African-American (2019–59.4, 2020–46.1, 2021–17.3) races as well (Wound area comparison over the visits and time period).

From Table 2, it can be seen that the wound area is relatively high in African-American patients for the years 2019 and 2020. The average wound area is increasing in Caucasian and Female patients from 2018 to 2021 as per the line chart using the program indicators by filtering races, gender, and average area.

Over the years of 2019–2021, we observed an increase in the wound area among Caucasian patients during their visits, while there was a significant decrease in the wound area during the visits of African-American patients. We can also see that the average wound area reduction in diabetic patients is slightly lower than in others. A significant difference is observed in wound healing status for diabetic patients. The wounds are healing in 4, 13, and 1 diabetic patients while not healing in 14, 13, and 3 diabetic patients for the third visit over the three years. Other parameters like hypertension, stroke, Peripheral Vascular Disease (PVD), and obesity can also be examined for the patients through event reports generation or appropriate program indicators. A growing pattern is observed between the Wound Area vs Glomerular Filtration Rate (GFR) value.

As we see, for every visit, patients having higher GFR value (>60) tended to have lower wound areas as seen from the scattered plot where the points are clustered there (Fig 3).

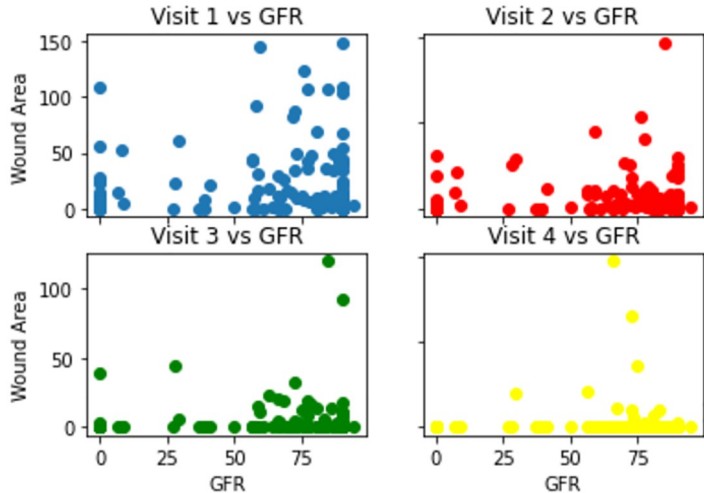

**Fig 3. Wound area vs GFR value.** The scatter plot of the Wound area Vs. GFR value for different visits.

## 3.3 Age distribution

The average age of the registered patients is 51–57. While the average age of the patients with healed wounds is 51–58, the average age of patients with non-healing wounds is 53–55 in 2019–2021.

## 3.4 County and zip-code distribution

Most of the registered patients with wounds are from Marion county (Fig 4). The population is dense in Marion county as per County Health Rankings Dataset (CHRD) [37] as well. Females are more from the areas 46214, 46201, 46203, 46205, and 46218 in Marion County and 46052 in Boone county. Most male patients are registered mainly from zip codes 46221, 46203, 46222, 46240, and 46256 within Marion county. The patients with non-healing wounds are from 46205, 46221, 46203, and 46214 in Marion county and 46052 from Boone county. The Caucasian population is the least in Marion county compared to others, while the African-Americans are more in Marion county, followed by Scott, Allen, and LaPorte.

At the final visit, the patients with non-healing wounds were mainly from Marion county (46205, 46221, 46203, 46241,46163, 46202, 46222), followed by Hancock, Parke, Johnson, Lawrence, and Montgomery counties. This data is retrieved using the event visualizer app through program indicators and selecting the relevant organization units (zip code and county). The Caucasian population is the least in Marion county compared to others, while the African-Americans are more in the same.

**3.4.1 Racial distribution in CHRD vs. REDcap.** We compared the patient distribution among the counties based on their race and found most of the registered Caucasian patients (125) in Marion county for CHRD while fewer patients in this county for REDCap. For RED-Cap, most Caucasian patients registered were from other counties surrounding Marion. On the other hand, an increased number of registered African-American patients were from Marion county for both REDCap and CHRD.

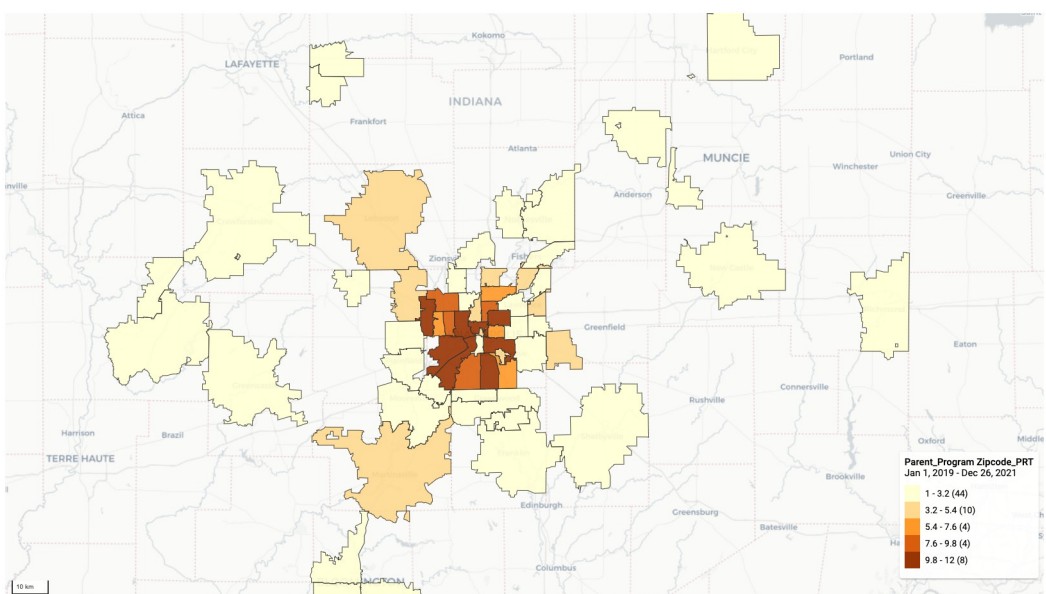

**Fig 4. Patient enrollment in different zip codes in Indiana.** The distribution of patient enrollment in different zip codes in Indiana from the interface of Map module in DHIS2.

### 3.5 Socio-economic conditions

The area-level socio-economic data (gender, race, poverty, unemployment, high school degree, income, and insurance) from the Polis center is in percentages ranging from 0–100%. Most of the patients are registered from the areas (Zip Code Tabulation Areas) where poverty (27.7%), unemployment (7.6%), no high school diploma (21.7%), and minority population (55.8%) rates are significantly higher than the other regions of the state. The total number of zip code tabulation areas in the system (Level 4 Organization Units) is 776. Among these, poverty is less than 20% in around 666 areas and 20–40% in 89 areas from the map application legend. We considered patients registered from the areas falling in the 20–40% category.

### 3.6 Comparison over the visits

Over the years of 2019, 2020, and 2021, we saw wound area increasing among Caucasian patients in all of their visits, while there was a significant decrease in wound area during the visits of African-American patients. We saw these interesting patterns from the instance's line chart form of visualization. The average wound area in visits 1, 2, and 3 for Caucasian patients was 32.7 sq. cm, 21.6 sq. cm & 15.6 sq. cm, and for African-American patients, they were 42.7 sq. cm, 32.1 sq. cm & 12.2 sq. cm (Fig 5).

Analyzing the patient visits from 2019 to 2021, we observed that during the initial visit, 84 patients exhibited wounds in the healing phase, while 103 patients presented with non-healing wounds. During the second visit, 112 patients demonstrated healing wounds, whereas 64 patients' wounds were classified as non-healing. By the third visit, 35 patients had healing wounds, and 37 patients' wounds were not healing. Utilizing the longitudinal visit data for each patient, we proceeded to develop forecasting models to predict wound healing status patterns and predict future wound conditions. However, the detailed forecasting modeling work is beyond the scope of this paper, which primarily focuses on describing the platform.

### 3.7 Socio-economic condition comparison over zip codes

Our analysis of the DHIS2 Maps application revealed that from 2019 to 2021, the majority of patients in our Parent Program for wound care resided in the following eight Indiana zip codes: 46234, 46222, 46241, 46221, 46202, 46218, 46203, and 46227. To control for potential

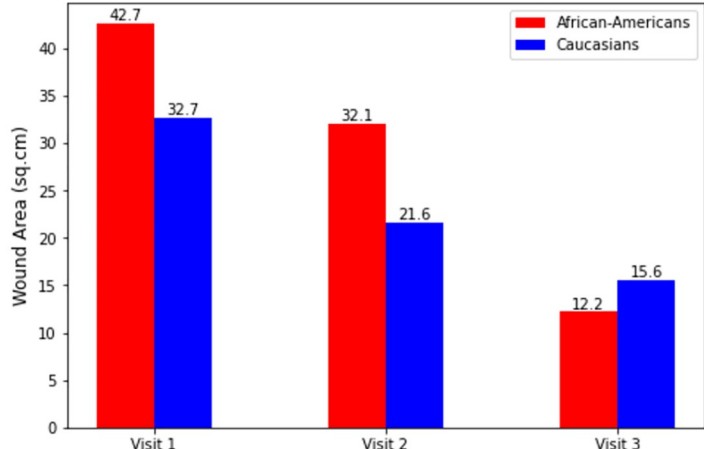

**Fig 5. Wound area over the visits for African-American and Caucasian patients.** The bar chart of mean wound area for African-American and Caucasian patients for different visits.

bias, we examined the demographic composition of these zip codes and found that Caucasian patients were primarily located in zip codes 46241, 46221, 46217, 46227, and 46203, while African-American patients were predominantly found in zip code 46222.

To further investigate potential confounding factors, we analyzed past employment history, current employment, education, and income rates among patients based on their residing zip codes. Our findings indicated that all zip codes with better past and current employment histories also exhibited higher income and education rates. Additionally, zip codes 46254, 46256, 46229, and 46163 demonstrated better current employment, income, and education rates, with the exception of 46254, which did not show higher income levels among patients. By considering these socio-economic factors and their distribution across zip codes, we aimed to minimize bias in our analysis and provide a more comprehensive understanding of the patient population. (Fig 6).

### 3.8 Usability feedback from researchers

**3.8.1 Internal validity score.** A purposive sampling method was employed to select 12 participants for the usability study of the DHIS2 platform, ensuring representation from various relevant user groups [38]. The sample included researchers engaged in laboratory (n = 5) or clinical research (n = 5) and medical or pre-med students (n = 2). This diverse sample was chosen to assess the platform's versatility and suitability for individuals at different stages of professional development, as the platform is intended to cater to a wide range of users within the healthcare and research domains. The primary inclusion criterion for all participants was proficiency in using a web browser on a computer, as this skill is essential for effectively navigating and utilizing the DHIS2 platform.

Prior to distributing DHIS2 platform credentials to the participants, the study protocol was reviewed and approved by the Institutional Review Board (IRB) of Indiana University, ensuring compliance with ethical guidelines. Each participant received comprehensive documentation to facilitate their understanding and use of the platform.

Due to the exploratory nature of this usability study and the focus on obtaining in-depth feedback from a diverse group of users, a sample size of 12 participants was deemed sufficient. This sample size allowed for the collection of detailed insights and experiences from each participant while maintaining a manageable scope for the study.

The participants answered 3 questions for each of the constructs from the UTAUT2 model and assisted us with their feedback to measure their performance & effort expectancy, facilitating conditions, hedonic motivation, social influence, and behavioral intention for using the developed DHIS2 instance in wound research. We found out the internal consistency of each of the UTAUT2 constructs from their survey questionnaire responses and measured it by the coefficient of reliability Cronbach's Alpha value. Tavakol et al. [39] discuss that Cronbach's alpha can be mainly used to measure the validity and reliability in the evaluation of a measurement instrument. Going through several studies [40–42], we decided to use Cronbach's alpha to evaluate the usability of our platform. The Cronbach's Alpha values for each of the constructs from the survey are shown in Table 3.

**3.8.2 Statistical tests.** We performed ANOVA and Tukey's Posthoc test to find a statistically significant difference in the responses among the three groups of participants in the survey study. The ANOVA test returned the F-statistic value as 4.7921 with a p-value of 0.009, which depicted statistically significant different responses among the three groups of Faculty/Researchers, Students, and Clinicians.

The participant responses in percentages for each statement is shown in Fig 7 and number of participants response in each statement and original survey data without identification are

(a)

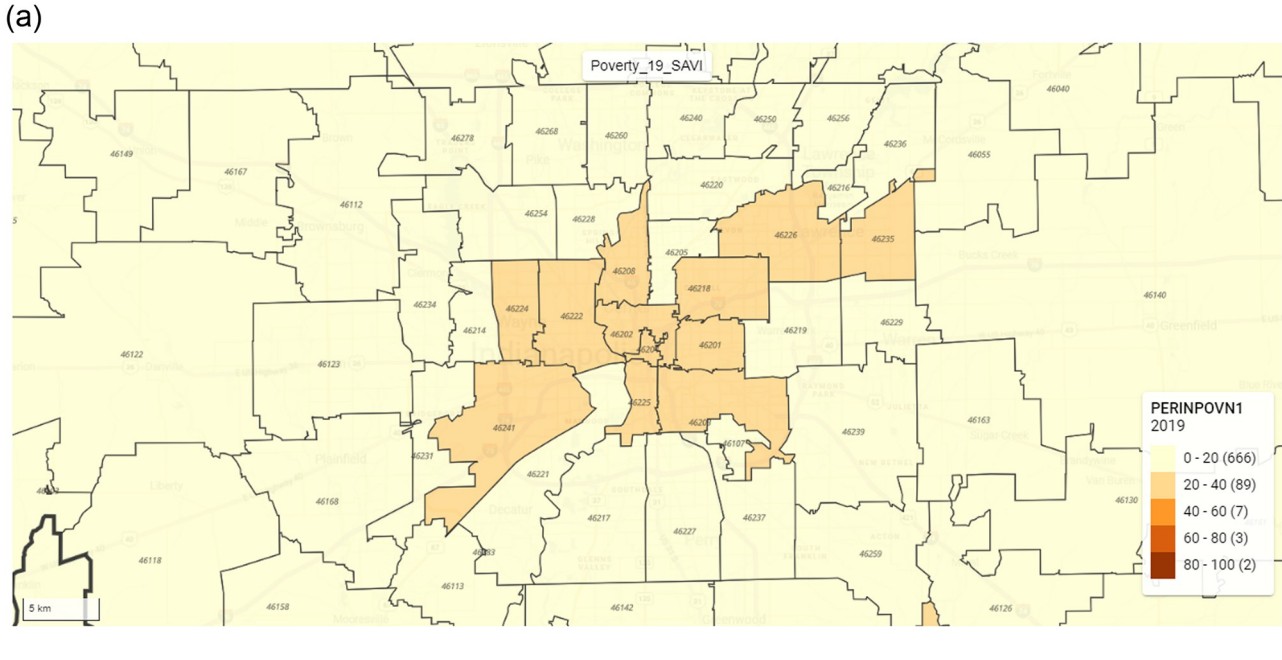

(b)

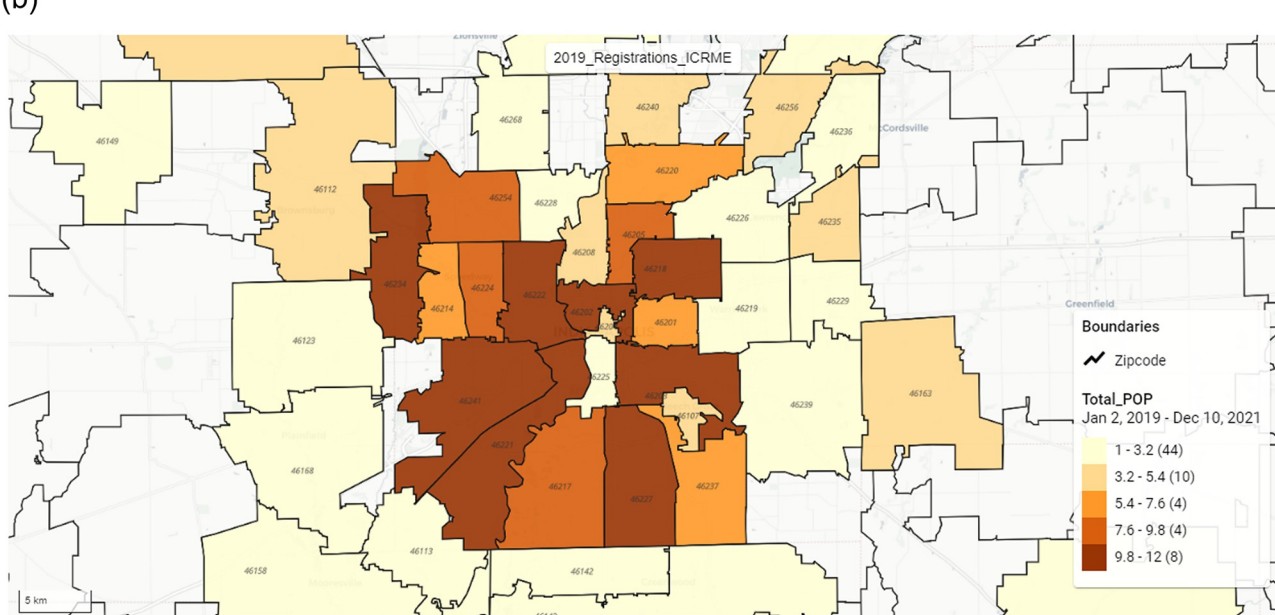

**Fig 6. Visualizations of zip codes having patients in 2019 in the wound studies at CWC.** (6a) Distribution of patients with poor financial condition in 2019. (6b) Distribution of all enrolled patients in 2019 in the wound research studies from the interface of Map module of DHIS2.

given in supplementary documents (S1 and S2 Files). To see which group(s) of participants showed statistically significant differences in their responses, we performed Tukey's posthoc test, and the test score is shown in Table 4. From this test, we saw a significant difference between the responses of the Faculty/Researchers group and the Student group, but we did not see it between Clinical Staffs group and either of the previously mentioned groups.

**Table 3. Internal consistency score.**

| Construct | Cronbach's $\alpha$ | 95% Confidence Interval |
|---|---|---|
| PE | 0.9336 | [0.824, 0.979] |
| EE | 0.8853 | [0.696, 0.964] |
| FC | 0.7001 | [0.206, 0.906] |
| HM | 0.7933 | [0.453, 0.935] |
| SI | 0.7804 | [0.419, 0.931] |
| BI | 0.6269 | [0.012, 0.883] |

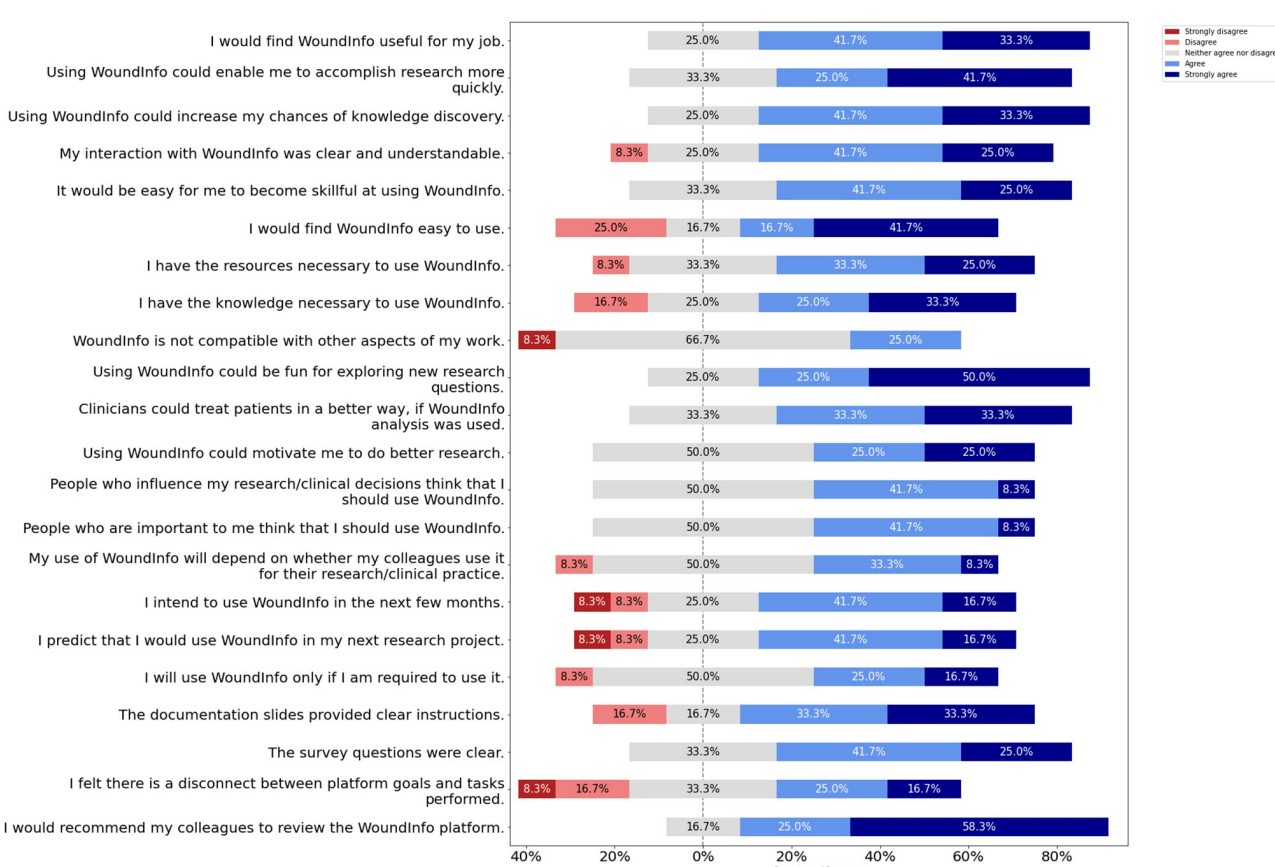

**Fig 7. Likert plot for participants response to each statement.** The percentage plot of the responses for each statement from the participants.

**Table 4. Tukey's posthoc test score.**

| Multiple Comparison of Means—Tukey HSD, FWER = 0.05 | | | | | | |
|---|---|---|---|---|---|---|
| Group 1 | Group 2 | Mean Difference | p-adj | Lower | Upper | Reject |
| Clinical Staffs | Faculties/Researchers | -0.1545 | 0.478 | -0.4685 | 0.1594 | False |
| Clinical Staffs | Students | 0.3909 | 0.0699 | -0.0244 | 0.8062 | False |
| Faculties/Researchers | Students | 0.5455 | 0.0061 | 0.1301 | 0.9608 | True |

## 4 Discussion

### 4.1 DHIS2 platform

The DHIS2 platform helped integrate socio-medical data from RedCAP and EHR and visualize it through user-friendly dashboards. On this platform, doctors can see from a patient's address if there is a pattern from a specific place the patient has come from, and they can plan the treatment or procedure. The visualization and combined report also give the clinician a preliminary idea of the patient's current condition and history. To generalize to other wound care centers across the nation, the parent program metadata can be easily shared so that we can generate a domain model for different patient characteristics.

From the registered patients' information imported into the instance, data visualization and distribution from the dashboard gave us precise observations. Access to several apps in the DHIS2 instance eased the process of comparing the programs. Compared to program management done in DHIS2 in previously published research, we merged multiple data formats—XML, CSV, JSON, etc. In several previous works for wound research mentioned in the literature review, healthcare-related data were recorded in other electronic record systems. However, this is one of the first kinds of analysis done on wound data in a platform like DHIS2, primarily based on social determinants of health. In [43], an article by Healogics Wound Science Initiatives, explains the number of social determinants the researchers have considered in the wound data analysis. While our work stayed in line with them, we aimed to visualize and quantify the actual social determinants as indicators of the wound healing status [44]. We also studied the need for electronic health records to move to value-based care for patients by paying for the performance of the physicians by Fife et al. [45],

We studied several works utilizing DHIS2 for large-scale health data integration. For example, Sahay et al. [46] explain that DHIS2, as an open-source platform, can be better than any other health management data system considering its advanced usability in aggregating data from many levels. Scott et al. [47] developed a bulletin through the DHIS2 platform to estimate and report Rabies data in African regions. In Botswana, researchers have worked on implementing the DHIS2 platform to convert the health record from paper to electronic system [48].

Some exploratory studies, such as work by Begum et al. [49], demonstrate that DHIS2 was adopted to utilize real-time health service data, which contributed to developing a responsive HMIS in countries where EHR systems were not in practice. However, entering real-time data into the system had been an issue in this work due to slow internet speeds and challenges with offline access. Serena et al. [50] shed light on a new approach to integrating clinical care and quality reporting based on several wound care clinics. The clinics or centers submitted their data to develop the common framework of EHR systems across a national network. The transformation of a national-level network of wound care clinics into DHIS2 has the potential for clinical effectiveness research. However, focusing only on clinical factors (which may be possible through a clinical registry or Health Information Exchange) might not reflect the true potential of a large-scale data warehouse like DHIS2 that can combine social, behavioral, and economic data with clinical data. Our work has integrated several wound care programs into one parent program at Indiana University Health (IUH). Using a national-level wound consortium platform such as Diabetic Foot Consortium, this can be extended to a national network as mentioned in [50]. For an integrated disease surveillance system, guidelines to create a national-level network without always relying upon DHIS2 can be found in [51], where it is under implementation in Nigeria.

Several notable works have looked at data integration through clinical registries. Fife et al. [52] developed clinical registries using DHIS2 to create a merit-based incentive payment

system. Russell et al. [53] describe the need for wound documentation and classification for better wound assessment but do not expand to its use in an electronic medium for the documentation. Our data import into the DHIS2 instance paves the way to get an appropriate domain model, similar to Sieben's domain models [54], but for the wound care clinical domain. Hess et al. [55] focused on documenting wound care information for a smart workflow synchronization with Electronic Health Records (EHR) in an outpatient wound care department. Karuri et al. [56] improved health data for some programs but did not include wounds or injury data in their use in Kenya. In contrast, Harrison et al. [57] discussed improving and advancing information on modern wound care and complex issues regarding clinical decision-making in wound care management, such as demographic changes, increases in obesity, alcoholic liver diseases, and complex surgeries associated with these. Using the DHIS2 platform can enhance their work's ease of implementation and generalizability. Flattau et al. [58] have used a national EHR network to describe wound data characteristics (sickle cell ulcer wound) and healing patterns, while Kaewprag et al. [59] developed predictive models for wounds (pressure ulcers) using EHR for ICU. Our work aligns with these, as we aim to expand our platform to the national level. Carter et al. [60] provide guidelines to create a healthcare database that can improve cohort matching, variable classification, and reporting for better wound care characterization.

Li et al. [61] evaluated the effectiveness of EHR in documenting pressure ulcers and found that well-documented EHR can lead to better health outcomes. By adding clinical documentation across multiple facilities, our work emphasizes improving wound treatment at the provider level. Wickström et al. [62] conducted a qualitative study on the effectiveness of a Digital Decision Support System (DSS) for wound management and identified four categories based on healthcare staff's experiences using it, suggesting that access to socio-economic data in the database could improve their experience.

## 4.2 Data analysis

Study by Fogerty et al. [63] observed increased risk among African-American wound (pressure ulcers) patients across all age groups, and in both males and females which is higher than the Caucasian wound patients. [64] also mentionec higher amuputation rate in diabetic African-American patients (twice of diabetic Caucasian patients). In our study, we clearly found the wound area relatively high in African-American patients in 2019 and 2020. A study by Bliss et al. [65] found a significantly smaller proportion of African-American patients' wound healed admitted in nursing homes compared to admitted Caucasian patients. We found the average wound area increasing in Caucasian and Female patients from 2018 to 2021 as per the line chart using the program indicators by filtering races, gender, and average area in our DHIS2 instance.

It is also found from the study that the patients having a non-healing status of the wounds in their several numbers of visits are more from the poor neighborhoods, areas with a high concentration of unemployment and minority population. While the racial distribution among the patients is not relatively equal, the patients are normally distributed in this category.

From the distribution of patients based on the zip codes or counties, we could see some of the zip codes or counties having more patients than most of the other areas, which implies that access to wound care centers is unequal. Our analytics platform can help identify the locations where the establishment of new wound care centers or outreach of existing wound care centers has to be done. Better resource planning can be done by adding more staff, doctors, and drug stocks to provide better treatment for the patients.

An interesting observation from the wound data for 2019 and 2020 is that most African-American patients have deeper wounds than Caucasian patients. Here we disregard the data of 2021 for not having all the data till December 2021. Some elements had missing data for the whole year of 2021. Thus, data from 2019 to 2021 is more valuable. For these two years, African-American patients' wound areas in all visits are significantly larger than Caucasian patients, implying African-American patients registered in specific zip codes can be thought of as more critical and should be prioritized in case of treatment, while establishing more wound centers in those areas. If IUH-CWC can act as a coordinator of the wound programs, it should also consider having more wound centers in those zip codes and more nurses and medical professionals assigned in those centers. The specification of zip codes or counties that need more importance can also be done clearly by looking at the forecasting models.

### 4.3 Future scope

Several wound management programs were integrated, establishing a balanced platform to manage integrated programs. There is further scope to categorize the type of wounds and their prevalence in the socio-economic groups. Our project used DHIS2 as a tool to map the data distribution of the patients based on demographics and used Polis Center data to check our mapping. To compare the distribution of the demographics, we are using RedCAP data. Importing every patient's data value into the system must be ensured to get a more accurate distribution of the demographics. We have followed the API external data importing system to post the data. However, the entire API data import process must be followed thoroughly to ensure that the instance has information from the programs for individual patients. We only observed racial distribution for a few races, which we can extend to all races and ethnicities in the instances dashboard, including all the programs and visualization in maps. In the future, the programs can be aggregated to inter-state and extended to the national level. Considering only one state, such as Indiana, as the data are coming from only a few wound centers, CWC can coordinate different wound centers in Indiana. To refer to the necessary and useful guidelines for building wound centers in the state, we can cite the work by Kim et al. [66] where the critical elements, such as the aspects of physical space, financial support, and building multi-disciplinary teams are discussed. Similar types of coordinators can be assigned for other states if data is aggregated from other states. Serena et al. [50] discussed a new approach of aggregating several wound care clinics and worked towards developing a common framework of EHR across a national network. DHIS2 could be a favorable medium to the discussed approach in the previous work to create a national network among wound care centers or clinics.

We can also consider the long-term impacts of implementing EHR for wound care in state-wise or country-level networks. Howley et al. [67] discussed the long-term financial impact of electronic health record implementation, which resulted in an increase in practice reimbursement and a decrease in patient visits. Getting the right platform like DHIS2 developed for wound care can better manage patient load, and with the appropriate analytics, the wound centers will be able to focus on the right patients. One of the applications and future steps of this project is to predict wound healing status using machine learning approaches, i.e., healed and unhealed wounds. Jung et al., 2016 [68] focused on similar research on the identification of slow-healing wounds.

Based on our findings from the DHIS2 WoundInfo pilot deployment, we can share interesting care protocol suggestions like the useful article by WoundsCanada [69], which describes best practices in developing a care plan for non-healing wounds. Similarly, our project can provide guidelines to wound centers that want to deploy EHRs or data warehouses, similar to the

work by Hess et al. [70] to improve patient care, increase patient safety and overall simplify wound care compliance in those centers.

DHIS2 allowed us to store the data on different levels, as organizational units gave us particular advantages over the other methods to aggregate and compare the wound data on zip code, county, or even state level. The scope of having multiple users on our platform gives us confidence that this deployment can go beyond a single health system Indiana University Health-Comprehensive Wound Center (IUH-CWC) to a multi-system, multi-state data analytics platform.

## 5 Conclusion

Understanding and monitoring patient clinical progress will pave the way to injury surveillance, as DHIS2 provides many options to customize and analyze patient information. Comparing the socio-economic characteristics of the patients and addressing the unique needs of the prevalent population leads to achieving effective treatment outcomes. DHIS2 can prove to be a good platform for generating reports and analyzing patient data at the provider level or in the wound centers. Such a surveillance mechanism in different facilities and at the state level will be critical to improving wound outcomes.

## Supporting information

**S1 File. Number of participants response to each statement.** Supplementary document 1 shows the number of responses for each statement from the participants who participated in the DHIS2 WoundInfo platform review study.
(PDF)

**S2 File. Survey data.** The CSV file contains the original survey data from the participants without their identification.
(CSV)

## Acknowledgments

We thank the Polis Center at IUPUI, REDCap, and WoundExpert team members at Indiana University for giving access to the data from the wound care centers. We thank the Indiana Center for Regenerative Medicine, and Engineering (ICRME) research team for their valuable suggestions and continuous feedback to improve the platform.

## Author Contributions

**Conceptualization:** Saptarshi Purkayastha, Chandan K. Sen.

**Data curation:** Atika Rahman Paddo, Snigdha Kodela, Saptarshi Purkayastha.

**Formal analysis:** Atika Rahman Paddo, Saptarshi Purkayastha.

**Funding acquisition:** Chandan K. Sen.

**Investigation:** Lava Timsina, Shomita S. Mathew-Steiner, Saptarshi Purkayastha, Chandan K. Sen.

**Methodology:** Atika Rahman Paddo, Saptarshi Purkayastha.

**Project administration:** Shomita S. Mathew-Steiner, Saptarshi Purkayastha, Chandan K. Sen.

**Resources:** Atika Rahman Paddo, Saptarshi Purkayastha.

**Software:** Atika Rahman Paddo, Snigdha Kodela, Saptarshi Purkayastha.

**Supervision:** Lava Timsina, Shomita S. Mathew-Steiner, Saptarshi Purkayastha, Chandan K. Sen.

**Validation:** Atika Rahman Paddo, Saptarshi Purkayastha.

**Visualization:** Atika Rahman Paddo, Saptarshi Purkayastha.

**Writing – original draft:** Atika Rahman Paddo, Saptarshi Purkayastha.

**Writing – review & editing:** Atika Rahman Paddo, Lava Timsina, Saptarshi Purkayastha, Chandan K. Sen.

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
