## [Decision Letter · Decision Letter 0]

21 Feb 2024

PONE-D-23-28893Development and Validation of the DHIS2 Platform for Integrating Sociomedical Data to Study Wound Care OutcomesPLOS ONE

Dear Dr. Paddo,

Thank you for submitting your manuscript to PLOS ONE. After careful consideration, we feel that it has merit but does not fully meet PLOS ONE’s publication criteria as it currently stands. Therefore, we invite you to submit a revised version of the manuscript that addresses the points raised during the review process.

**ACADEMIC EDITOR: ** 

Dear Author,

Please revise all aspects highlighted by the reviewer, particularly focusing on the statistical analysis of your manuscript. Our decision is guided by PLOS ONE’s publication criteria.

Best regards,

José Luiz Vieira

We look forward to receiving your revised manuscript.

Kind regards,

José Luiz Fernandes Vieira

Academic Editor

PLOS ONE

Journal Requirements:

3. We note that Figures 4 and 6 in your submission contain [map/satellite] images which may be copyrighted. All PLOS content is published under the Creative Commons Attribution License (CC BY 4.0), which means that the manuscript, images, and Supporting Information files will be freely available online, and any third party is permitted to access, download, copy, distribute, and use these materials in any way, even commercially, with proper attribution. For these reasons, we cannot publish previously copyrighted maps or satellite images created using proprietary data, such as Google software (Google Maps, Street View, and Earth). For more information, see our copyright guidelines: http://journals.plos.org/plosone/s/licenses-and-copyright.

a. You may seek permission from the original copyright holder of Figures 4 and 6to publish the content specifically under the CC BY 4.0 license.  

Reviewers' comments:

Reviewer's Responses to Questions

**Comments to the Author**

1. Is the manuscript technically sound, and do the data support the conclusions?

Reviewer #1: Yes

2. Has the statistical analysis been performed appropriately and rigorously? 

Reviewer #1: Yes

3. Have the authors made all data underlying the findings in their manuscript fully available?

Reviewer #1: No

4. Is the manuscript presented in an intelligible fashion and written in standard English?

Reviewer #1: Yes

5. Review Comments to the Author

Reviewer #1: The study is well executed and provides interesting insights regarding the capabilities of DHIS2.

I missed the study objectives. However, I could decipher the intent of the study from explanations provided. It would be better to make the objectives explicit, to enable reviewers to assess key results effectively. I also missed descriptive statistics as indicated in the text.

It would also be beneficial to describe the limitations of the study, taking into consideration sources of potential bias, e.g. other data that were not used - type of health services clients accessed, co-morbidities.

6. PLOS authors have the option to publish the peer review history of their article (what does this mean?). If published, this will include your full peer review and any attached files.

Reviewer #1: No

---

## [Author Response · Author response to Decision Letter 0]

26 Mar 2024

Thank you for the opportunity to revise and resubmit our manuscript titled “Development and Validation of the DHIS2 Platform for Integrating Sociomedical Data to Study Wound Care Outcomes” (Manuscript ID: PONE-D-23-28893) for further consideration in PLOS One. We greatly appreciate the time and effort from the reviewers and you, in providing valuable feedback to improve our work.

We have carefully addressed all the comments and concerns raised by the reviewers. We have made the necessary revisions throughout the manuscript, which are highlighted for ease of review. The major revisions addressing the comments from the reviewers and the editor are as follows:

Comments from the reviewer:

1. Comment 1: Maybe an acknowledgement from authors that these were the objectives for this study?

Response: We changed the section (pg. 2, paragraph 3) to make it study objective.

2. Comment 2: Further clarification is required regarding specific clinical trial data used, stage of clinical trial. Measures to address confounding.

Response: We changed the paragraph to highlight the clinical studies whose data was imported into the platform as follows: The incorporated clinical studies are registered on ClinicalTrials.gov under the following identifiers: NCT02577120, NCT02581098, NCT03793062, and NCT01101854.

3. Comment 3: This sounds incomplete.

Response: We rephrased the sentence to add more clarity (Section 2.1, pg. 3).

4. Comment 4: I could not find the predictive results.

Response: The detailed forecasting modeling work is beyond the scope of this paper which primarily focuses on the platform and its evaluation.

5. Comment 5: How was bias controlled?

Response: We re-worded the paragraph 3.7 to discuss bias control as follows:

To control for potential bias, we examined the demographic composition of these zip codes and found that Caucasian patients were primarily located in zip codes 46241, 46221, 46217, 46227, and 46203, while African-American patients were predominantly found in zip code 46222.

To further investigate potential confounding factors, we analyzed past employment history, current employment, education, and income rates among patients based on their residing zip codes. Our findings indicated that all zip codes with better past and current employment histories also exhibited higher income and education rates. Additionally, zip codes 46254, 46256, 46229, and 46163 demonstrated better current employment, income, and education rates, with the exception of 46254, which did not show higher income levels among patients. By considering these socio-economic factors and their distribution across zip codes, we aimed to minimize bias in our analysis and provide a more comprehensive understanding of the patient population.

6. Comment 6: Not enough detail about eligibility criteria and methods of selecting the sample. What were the reasons for recruiting only 12 participants for such an important innovation?

Response: The expected criteria for the inclusion of participants for the study has been re-written and included in the manuscript (Section 3.8.1, pg. 9). The reason to choose the number of participants is also explained.

7. Comment 7: It would be helpful to add the results based on their responses to assessed constructs. These results show group differences only.

Response: We added the participant responses in a Table in the supplementary document. We also added a Figure representing the participant responses.

8. Comment 8: This is indicative of future plans, contrary to the statement above, that predictive models were built.

Response: We rephrased this line to add more clarity that forecasting models to predict healing trajectories and classification models to predict wound healing status are out of scope for this paper (pg. 11, paragraph 2).

9. Comment 9: The researchers were able to provide evidence to this claim. A good pilot study.

Response: Thank you for the comment. It is indeed a pilot study to judge the feasibility of the WoundInfo platform.

10. Comment 10: This would be a useful initiative in contexts where standards for data management exist. This would reduce risk of bias.

Response: Thank you for the comment.

Response for additional comments from the editor:

11. Thank you for your inquiry regarding the copyright permissions for the figures included in our submitted manuscript. Upon careful review, we have determined that the figures do not require a

separate Copyright form. The figures in question were generated using data from the WoundInfo DHIS2 instance, which utilizes the OpenStreetMap layer. OpenStreetMap is licensed under the Open Data Commons Open Database License (ODbL), which allows for the use and reproduction of the data for various purposes, including academic research. The process for creating these figures involved importing the necessary organization units into the DHIS2 instance. The map figures were then generated directly from the built-in map app module within the DHIS2 platform. As a result, the figures are considered original visualizations created by the authors using the available tools and data within the DHIS2 system.

We firmly believe that the revised manuscript has been substantially improved and addresses all the concerns raised during the previous review cycle. The added clarifications, expanded discussions, and additional analyses have strengthened the study's scientific merit and potential impact.

Thank you again for your consideration, and we look forward to hearing from you regarding the outcome of this revised submission.

---

## [Decision Letter · Decision Letter 1]

26 Jul 2024

Development and Validation of the DHIS2 Platform for Integrating Sociomedical Data to Study Wound Care Outcomes

PONE-D-23-28893R1

Dear Dr. Pado

We’re pleased to inform you that your manuscript has been judged scientifically suitable for publication and will be formally accepted for publication once it meets all outstanding technical requirements.

Kind regards,

José Luiz Fernandes Vieira

Academic Editor

PLOS ONE

Additional Editor Comments (optional):

All the comments were ansewers by the authors.

best regards

José Luiz vieira

Reviewers' comments:

Reviewer's Responses to Questions

**Comments to the Author**

1. If the authors have adequately addressed your comments raised in a previous round of review and you feel that this manuscript is now acceptable for publication, you may indicate that here to bypass the “Comments to the Author” section, enter your conflict of interest statement in the “Confidential to Editor” section, and submit your "Accept" recommendation.

Reviewer #1: All comments have been addressed

2. Is the manuscript technically sound, and do the data support the conclusions?

Reviewer #1: Yes

3. Has the statistical analysis been performed appropriately and rigorously? 

Reviewer #1: Yes

4. Have the authors made all data underlying the findings in their manuscript fully available?

Reviewer #1: Yes

5. Is the manuscript presented in an intelligible fashion and written in standard English?

Reviewer #1: Yes

6. Review Comments to the Author

Reviewer #1: The authors have adequately addressed the comments. Additional explanation provided to all areas. There is no potential risk for conflict of interest, research and publication ethics.

7. PLOS authors have the option to publish the peer review history of their article (what does this mean?). If published, this will include your full peer review and any attached files.

Reviewer #1: No

---

## [Editor Report · Acceptance letter]

8 Aug 2024

PONE-D-23-28893R1 

PLOS ONE

Dear Dr. Paddo, 

I'm pleased to inform you that your manuscript has been deemed suitable for publication in PLOS ONE. Congratulations! Your manuscript is now being handed over to our production team.

Kind regards, 

on behalf of

Dr. José Luiz Fernandes Vieira 

Academic Editor

PLOS ONE